# Lack of association between loneliness, social isolation and inflammation in people living with HIV aged ≥50 years: Results from the sub-study "No One Alone-Gesida Study"

Jose-Ramon Blanco[1,2]*, Helena Albendin-Iglesias[3], Eugenia Negredo-Puigmal[4], Ana Mª Barrios-Blandino[5], Cristina Tomás-Jimenez[6], Isabel Sanjoaquin-Conde[7], María Saumoy[8], Verónica Pérez-Esquerdo[9], Inmaculada Gonzalez-Cuello[10], Ana María López-Lirola[11], María José Galindo[12], Noemí Cabello-Clotet[13], Jesica Abadía Otero[14], Dolores Merino-Muñoz[15], Joanna Cano-Smith[16], Magdalena Muelas-Fernandez[17], Javier De La Torre[18], Alicia González-Baeza[19], Lourdes Romero[2], Antonio Ocampo[20], Rafael Torres[21], Carmen Hidalgo[22], Herminia Esteban[23], Maria Angeles Fernandes-López[3], Jordi Puig[4], Lucio Garcia Fraile[5], Enrique Bernal Morell[6], Laura Perez-Martinez[2], Marta De Miguel Montero[23], Inma Jarrin[24], Julian Olalla[18], THE GESIDA 12021 STUDY GROUP¶

1 Hospital Universitario San Pedro, Logroño, Spain, 2 Centro de Investigación Biomédico de La Rioja, Logroño, Spain, 3 Hospital Virgen de la Arrixaca, Murcia, Spain, 4 Hospital Universitari Germans Trias i Pujol, Badalona, Spain, 5 Hospital Universitario de La Princesa, Madrid – Centro de Investigación Biomédica en Red de Enfermedades Infecciosas (CIBERINFEC), Instituto de Salud Carlos III, Madrid, Spain, 6 Hospital General Universitario Reina Sofía, Murcia, Spain, 7 Hospital Clínico Universitario Lozano Blesa, Zaragoza, Spain, 8 Hospital Universitari de Bellvitge, L´Hospitalet de Llobregat, Spain, 9 Instituto de Investigación Sanitaria y Biomédica de Alicante, Alicante, Spain, 10 Hospital Vega Baja Orihuela, Alicante, Spain, 11 Hospital Universitario de Canarias, Canarias, Spain, 12 Hospital Clínico de Valencia, Valencia, Spain, 13 Hospital Clínico San Carlos, Madrid, Spain, 14 Hospital Rio Hortega, Valladolid, Spain, 15 Hospital Juan Ramón Jiménez, Huelva, Spain, 16 Hospital Universitario La Paz, Madrid, Spain, 17 Hospital de Viladecans, Viladecans, Spain, 18 Hospital Costa del Sol, Marbella, Spain, 19 Universidad Autónoma de Madrid, Madrid, Spain, 20 Hospital Álvaro Cunqueiro, Vigo, Spain, 21 Hospital Severo Ochoa, Leganes, Spain, 22 Hospital Virgen de las Nieves, Granada, Spain, 23 Fundación SEIMC-GESIDA, Madrid, Spain, 24 National Center for Epidemiology, Institute of Health Carlos III (ISCIII), Madrid, Spain. Center of Biomedical Research for Infectious Diseases (CIBERINFEC), Institute of Health Carlos III (ISCIII), Madrid, Grupo de Estudio Gesida, Spain

¶ Membership of the GESIDA 12021 Study Group is provided in the Acknowledgments.
* jrblancoramos@gmail.com

## Abstract

### Introduction

Inflammation is linked to multiple health conditions. Emerging evidence suggest it may play a role in the association between social isolation – loneliness and health outcomes. People living with HIV (PLWH) exhibit chronic inflammation even with viral suppression, likely due to ongoing immune activation. However, few studies have explored inflammatory biomarker in PLWH experiencing loneliness or social isolation. This study aimed to assess this association in PLWH aged 50 or older.

**Data availability statement:** Individual-level data from human participants cannot be made publicly available due to privacy and informed-consent restrictions. De-identified data will be available to qualified researchers upon reasonable request and completion of a data-use agreement, subject to approval by the Institutional Research Ethics Committee (Comité de Ética de Investigación con medicamentos de La Rioja [CEImLAR]) from the CEIC secretary: secretaria.ceic@riojasalud.es.

**Funding:** This study was supported by Gilead Science in the form of a grant awarded to J.R.B. (ISRES-18-10529) and Gilead Fellowship for Biomedical Research to J.R.B. (Beca Gilead a la Investigación Biomédica). The specific roles of this author are articulated in the 'author contributions' section. The funders had no role in study design, data collection and analysis, decision to publish, or preparation of the manuscript.

**Competing interests:** NO authors have competing interests in this article.

## Materials and methods

This observational, multicenter, cross-sectional study was conducted by the AIDS Study Group of the Spanish Society of Infectious Diseases and Clinical Microbiology across 22 Spanish hospitals. Participants were classified based on whether they experienced loneliness (UCLA 3-item scale ≥6) and/or social isolation (Lubben Social Network Scale-6 ≤ 20). Blood samples were stored at −80°C, and cytokine levels were measured using the Olink Target 48 Cytokine panel. Multivariable median regression was used to assess differences in inflammatory biomarkers between those experiencing loneliness and/or social isolation and those experiencing neither.

## Results

Among 199 PLWH, 33.2% reported loneliness/social isolation. Those without loneliness/social isolation were more likely to be male (p = 0.023), employed or students (p < 0.001) and had fewer prior AIDS events (p = 0.045) and comorbidities. CSF1 levels were elevated in unadjusted analysis (median difference 11.33; 95% CI 3.50–19.15; p = 0.005) but not in adjusted models.

## Conclusion

No significant increase in inflammatory markers was found in PLWH experiencing loneliness and/or social isolation. Further research should prioritize longitudinal, multi–time-point designs with expanded biomarker panels and test whether connection-enhancing interventions modulate inflammation.

## Introduction

Despite the success of antiretroviral therapy (ART), people living with HIV (PLWH) are at higher risk of noncommunicable diseases than age-matched HIV-negative individuals. The underlying causes of this elevated risk are not fully understood, though they may involve an accelerated or accentuated aging process [1–3]. This process is likely driven by a complex interplay of factors, including HIV infection, ART, chronic viral co-infections, and lifestyle factors.

Until recently, health has been characterized by eight biological hallmarks that influence overall health [4]. These hallmarks affect various biological system across the body. Recently, however, mental and socioeconomic factors have been proposed as additional critical hallmark of health [5]. In this context, the impact of mental state on biological aging is considered substantial, potentially exerting effects comparable to those of smoking [6].

Social isolation (SIL) and loneliness significantly impact the well-being, mental health, and quality of life (QoL) of PLWH [7,8]. SIL refers to objectively limited social contact, whereas loneliness reflects the subjective distress arising from a discrepancy between desired and actual social relationships [9]. Although these phenomena often co-occur, they are distinct: individuals may experience loneliness despite frequent social interactions [10]. Both have been linked to adverse health outcomes, including increased all-cause mortality, unhealthy behaviours, and mental health issues [11–16], with particularly strong impacts in older PLWH, who experience them more often than the general population [15,17].

Social disconnection triggers a biological stress response that disrupts endocrine, immune, metabolic, and cardiovascular systems and the gut–microbiome interface, with downstream effects on cortisol regulation, blood pressure, host defenses, and inflammatory activity [18]. SIL acts as a chronic stressor that increases mortality risk and the incidence of mental and cardiometabolic disorders through hyperactivation of the hypothalamic–pituitary–adrenal axis, glucocorticoid resistance, and inflammation; mitochondrial dysfunction serves as a central immunometabolic hub linking stress to disease. These changes promote depression, anxiety, and metabolic syndrome and, in turn, amplify social disconnection, reinforcing a self-perpetuating cycle [19,20]. Inflammation is a common pathway through which stress contribute to many chronic conditions. Evidence across disciplines implicates inflammatory process in a broad spectrum of physical and mental health outcomes [21,22], and accumulating data indicate that inflammation mediates, at least in part, the health effects of loneliness [23–27]. PLWH exhibit persistently elevated systemic inflammation compared to HIV-negative individuals, even with virological suppression, consistent with ongoing immune activation. [28,29].

SIL and loneliness are prevalent among PLWH and are associated with heightened inflammation [24,30,31], a key driver of HIV-associated comorbidities [32]. Although distinct [9], SIL and loneliness frequently co-occur [10] and may synergistically exacerbate inflammatory responses in PLWH. We conducted a cross-sectional study to evaluate whether loneliness, SIL, or their combination were associated with inflammatory markers in PLWH, hypothesizing that that reporting loneliness and/or SIL would exhibit higher concentrations of selected inflammatory markers compared to those without these experiences.

## Materials and methods

### Ethics statement

This study was approved by the Institutional Research Ethics Committee (Comité de Ética de Investigación con medicamentos de La Rioja [CEImLAR]). All participants provided written informed consent.

### Study design and participants

The study design has been described previously [33,34]. Briefly, this was an observational, cross-sectional, multicenter study conducted in 22 Spanish hospitals (September 2022 – May 2023. Eligible participants were PLWH aged ≥50 years in active follow-up at participating centers; those with a life expectancy <1 year were excluded at the treating physician's discretion.

Questionnaires and venous blood sampling were conducted as part of the same study. Participants provided serum samples for biomarker analyses and gave written informed consent before any procedures.

### Data collection

Data were obtained via self-report (face-to-face interviews and self-administered questionnaires) and systematic chart review, and captured in REDCap. From records we abstracted sociodemographic characteristics (age, sex at birth, place of birth/residence, education, employment, personal and marital status) and clinical history. HIV-related variables included duration of infection, route of acquisition, prior AIDS-defining events, CD4 nadir, current CD4 count, HIV RNA (undetectable <50 copies/mL), HBV coinfection (HBsAg+) and HCV coinfection (RNA+). Adherence in the previous 4 weeks was self-report. Comorbidities across major organ systems were recorded [35]. Polypharmacy was defined as >5 non-ART medications.

### Loneliness and social isolation

Loneliness was assessed with the 3-item UCLA Loneliness Scale; scores ≥6 classified participants as lonely [36]. Social isolation was assessed with the Lubben Social Network Scale–Revised (LSNS-R); scores ≤20 classified participants as socially isolated [37,38].

These scales were administered in their validated Spanish versions: UCLA [39] and Lubben [40]. Internal consistency for this cohort was previously reported (Cronbach's α = UCLA 0.896; and Lubben 0.842 [33].

## Psychosocial measures

Anxiety and depressive symptoms were assessed with the Hospital Anxiety and Depression Scale (HADS) (≥8 indicates elevated symptoms) [41]; HIV-related stigma with the Spanish-adapted HIV Stigma Scale (higher scores indicate greater stigma) [42]; and health-related quality of life with the EQ-5D-5L plus the EQ-5D visual analogue scale (higher scores indicate better health) [43,44]. Alcohol use was assessed with AUDIT-C (cut-offs >4 in men, > 3 in women) [45,46]. Tobacco and illicit drug use were self-reported.

These scales were administered in their validated Spanish versions: HADS and EQ-5D. Internal consistency for this cohort was previously reported (Cronbach's α = HADS 0.896; EQ-5D 0.795) [33].

## Inflammation biomarkers

Serum samples were obtained from blood drawn during a study visit and stored at −80° C. To assess inflammatory cytokine levels, the Olink Target 48 Cytokine panel was selected (https://olink.com/products/olink-target-48). Detailed information about the panel can be found in the S1 Table.

## Statistical Analysis

Participants were classified based on the presence of loneliness (UCLA-3 scale ≥6) and/or social isolation (LUBBEN scale ≤20). Descriptive analysis included frequency tables for categorical variables, and mean and standard deviation (SD) or median and interquartile range (IQR) for quantitative variables with normal or non-normal distribution, respectively. Differences in participant characteristics by loneliness and/or social isolation status were assessed using chi-square test for categorical variables and either the Student t-test or the non-parametric Mann-Whitney U test for continuous variables with normal or non-normal distribution, respectively.

At the analyte level, we required ≥70% detectability. Inflammatory proteins with >30% of measurements below the assay-specific lower limit of detection (LOD) were excluded (IL-33, IL-2, IL-1β, IL-4, TSLP, IL-13, and CSF2) [47]. For retained biomarkers, sub-LOD values were set to the LOD; this applied to IL-17A and IL-17F. Given the skewed distribution of inflammatory markers, we calculated the medians (IQR) of markers by loneliness and/or social isolation status, and used quantile regression models to assess differences between individuals experiencing loneliness and/or social isolation and those without it. Multivariable models were adjusted for the following variables, which are commonly associated with inflammatory markers in PLWH. Briefly, demographic factors (age, sex) [48], HIV-related immune status (CD4/CD8 ratio, viral suppression, duration of infection) [49,50], health status (multimorbidity, polypharmacy) [51], behaviors (tobacco use) [52], mental health (clinically significant anxiety/depression) [53], and socioeconomic context (employment status) [54] have been associated with systemic inflammatory biomarkers in PLWH.

All statistical analysis were conducted using Stata software (version 17.0; Stata Corporation, College Station, TX, USA).

## Results

A total of 199 PLWH were included in this analysis: 66.8% reported neither loneliness nor SIL, 13.6% loneliness only, 9.5% SIL only, and 10.1% both.

Loneliness and SIL were first analyzed as separate exposures, with participants initially classified into four mutually exclusive categories: neither loneliness nor social isolation, loneliness only, social isolation only, or both. Because some strata were small and no meaningful differences in inflammatory markers were observed among the loneliness only, social isolation only, and both groups, we combined these categories into a single exposure (loneliness and/or social isolation) to improve precision.

Socio-demographic data are shown in **Table 1**. Individuals experiencing loneliness and/or SIL were significantly more likely to be female (p = 0.023), unemployed/retired (p < 0.001), and living alone unwillingly (p = 0.006).

HIV-related data (**Table 2**) show that individuals without loneliness and/or social isolation had significantly fewer prior AIDS events (p = 0.045).

Comorbidity data (**Table 3**) reveal that those without loneliness and/or social isolation had significantly fewer musculo-skeletal (p < 0.001) and neuro-psychiatric (p = 0.02) comorbidities, as well as lower rates of multimorbidity (p = 0.001) and polypharmacy (p = 0.002).

Regarding tobacco, alcohol, and drug use (**Table 4**), no significant differences were observed between groups.

**Table 1. Characteristics of 199 people with HIV included in the study.**

|  | Loneliness and/or social isolation | | | |
|  | No | Yes | p-value | Total |
|  | N = 133 | N = 66 |  | N = 199 |
| Sex at birth [N (%)] |  |  | **0.023** |  |
| Male | 103 (77.4) | 41 (62.1) |  | 144 (72.4) |
| Female | 30 (22.6) | 25 (37.9) |  | 55 (27.6) |
| Age, years, mean (SD) | 59.1 (5.6) | 60.5 (6.2) | 0.212 | 59.6 (5.8) |
| Age, years, [N (%)] |  |  | 0.582 |  |
| 50-59 | 74 (55.6) | 34 (51.5) |  | 108 (54.3) |
| 60+ | 59 (44.4) | 32 (48.5) |  | 91 (45.7) |
| Race [N (%)] |  |  | 0.373 |  |
| Caucasian | 127 (95.5) | 61 (92.4) |  | 188 (94.5) |
| Other | 6 (4.5) | 5 (7.6) |  | 11 (5.5) |
| Place of birth [N (%)] |  |  | 0.538 |  |
| Spain | 117 (88.0) | 56 (84.8) |  | 173 (86.9) |
| Outside Spain | 16 (12.0) | 10 (15.2) |  | 26 (13.1) |
| Place of residence [N (%)] |  |  | 0.369 |  |
| Urban | 117 (88.0) | 55 (83.3) |  | 172 (86.4) |
| Rural (< 10.000 inhabitants) | 16 (12.0) | 11 (16.7) |  | 27 (13.6) |
| Education level [N (%)] |  |  | 0.078 |  |
| Elementary or lower | 38 (28.6) | 29 (43.9) |  | 67 (33.7) |
| Secondary or higher | 77 (57.9) | 28 (42.4) |  | 105 (52.8) |
| Unknown | 18 (13.5) | 9 (13.6) |  | 27 (13.6) |
| Employment status [N (%)] |  |  | **<0.001** |  |
| Student/Employed | 80 (60.2) | 19 (28.8) |  | 99 (49.7) |
| Retired/Unemployed | 48 (36.1) | 45 (68.2) |  | 93 (46.7) |
| Unknown | 5 (3.8) | 2 (3.0) |  | 7 (3.5) |
| Living situation, [N (%)] |  |  | **0.006** |  |
| Not living alone/ Lives alone by choice | 111 (83.5) | 50 (75.8) |  | 161 (80.9) |
| Lives alone unwillingly | 3 (2.3) | 9 (13.6) |  | 12 (6.0) |
| Unknown | 19 (14.3) | 7 (10.6) |  | 26 (13.1) |
| Marital Status [N (%)] |  |  | 0.663 |  |
| Married/Partnered/Separated/Widowed | 77 (57.9) | 34 (51.5) |  | 111 (55.8) |
| Single | 44 (33.1) | 26 (39.4) |  | 70 (35.2) |
| Unknown | 12 (9.0) | 6 (9.1) |  | 18 (9.0) |

**Table 2. Data related to HIV and coinfections, as a function of loneliness and/or social isolation.**

| | Loneliness and/or social isolation | | | |
| --- | --- | --- | --- | --- |
| | No | Yes | p-value | Total |
| | N = 133 | N = 66 | | N = 199 |
| Time since HIV diagnosis, years, [N (%)] | | | 0.712 | |
| <10 | 22 (16.5) | 8 (12.1) | | 30 (15.1) |
| 10-19 | 39 (29.3) | 20 (30.3) | | 59 (29.6) |
| ≥20 | 72 (54.1) | 38 (57.6) | | 110 (55.3) |
| HIV acquisition route, [N (%)] | | | 0.583 | |
| Heterosexual | 43 (32.3) | 20 (30.3) | | 63 (31.7) |
| Men who have sex with men | 54 (40.6) | 22 (33.3) | | 76 (38.2) |
| Intravenous drug users | 23 (17.3) | 17 (25.8) | | 40 (20.1) |
| Other | 12 (9.0) | 7 (10.6) | | 19 (9.5) |
| Unknown | 1 (0.8) | 0 (0.0) | | 1 (0.5) |
| CD4 nadir, mm$^3$, [N (%)] | | | 0.330 | |
| <200 | 57 (42.9) | 27 (40.9) | | 84 (42.2) |
| ≥200 | 72 (54.1) | 39 (59.1) | | 111 (55.8) |
| Unknown | 4 (3.0) | 0 (0.0) | | 4 (2.0) |
| Current CD4 count, mm$^3$, [N (%)] | | | 0.791 | |
| <500 | 30 (22.6) | 16 (24.2) | | 46 (23.1) |
| ≥500 | 103 (77.4) | 50 (75.8) | | 153 (76.9) |
| Current CD4/CD8 ratio, [N (%)] | | | 0.118 | |
| ≤1 | 80 (60.2) | 32 (48.5) | | 112 (56.3) |
| >1 | 53 (39.8) | 34 (51.5) | | 87 (43.7) |
| Current HIV RNA<50 copies/mL, [N (%)] | | | 0.090 | |
| No | 13 (9.8) | 2 (3.0) | | 15 (7.5) |
| Yes | 120 (90.2) | 64 (97.0) | | 184 (92.5) |
| Prior AIDS event, [N (%)] | | | **0.045** | |
| No | 106 (79.7) | 44 (66.7) | | 150 (75.4) |
| Yes | 27 (20.3) | 22 (33.3) | | 49 (24.6) |
| Lipoatrophy, [N (%)] | | | 0.787 | |
| No | 105 (78.9) | 51 (77.3) | | 156 (78.4) |
| Yes | 28 (21.1) | 15 (22.7) | | 43 (21.6) |
| Lipohypertrophy, [N (%)] | | | 0.518 | |
| No | 119 (89.5) | 57 (86.4) | | 176 (88.4) |
| Yes | 14 (10.5) | 9 (13.6) | | 23 (11.6) |
| Hepatitis B coinfection, (HBsAg+) [N (%)] | | | 0.617 | |
| No | 127 (95.5) | 64 (97.0) | | 191 (96.0) |
| Yes | 6 (4.5) | 2 (3.0) | | 8 (4.0) |
| Hepatitis C coinfection (ARN+), [N (%)] | | | 0.259 | |
| No | 97 (72.9) | 43 (65.2) | | 140 (70.4) |
| Previous (cured) | 34 (25.6) | 23 (34.8) | | 57 (28.6) |
| Active | 2 (1.5) | 0 (0.0) | | 2 (1.0) |
| ART Adherence in the last 4 weeks, [N (%)] | | | 0.530 | |
| >95 | 126 (94.7) | 61 (92.4) | | 187 (94.0) |
| <95 | 6 (4.5) | 5 (7.6) | | 11 (5.5) |
| Unknown | 1 (0.8) | 0 (0.0) | | 1 (0.5) |

**Table 3. Comorbidities, as a function of loneliness and/or social isolation.**

| | Loneliness and/or social isolation | | | Total |
|---|---|---|---|---|
| | No | Yes | p-value | |
| | N = 133 | N = 66 | | N = 199 |
| Cardiovascular, [N (%)] | | | 0.449 | |
| No | 82 (61.7) | 37 (56.1) | | 119 (59.8) |
| Yes | 51 (38.3) | 29 (43.9) | | 80 (40.2) |
| Digestive, [N (%)] | | | 0.165 | |
| No | 126 (94.7) | 59 (89.4) | | 185 (93.0) |
| Yes | 7 (5.3) | 7 (10.6) | | 14 (7.0) |
| Endocrine, [N (%)] | | | 0.885 | |
| No | 78 (58.6) | 38 (57.6) | | 116 (58.3) |
| Yes | 55 (41.4) | 28 (42.4) | | 83 (41.7) |
| Respiratory, [N (%)] | | | 0.194 | |
| No | 121 (91.0) | 56 (84.8) | | 177 (88.9) |
| Yes | 12 (9.0) | 10 (15.2) | | 22 (11.1) |
| Musculoskeletal, [N (%)] | | | **<0.001** | |
| No | 115 (86.5) | 42 (63.6) | | 157 (78.9) |
| Yes | 18 (13.5) | 24 (36.4) | | 42 (21.1) |
| Neuro-psychiatric, [N (%)] | | | **0.020** | |
| No | 100 (75.2) | 39 (59.1) | | 139 (69.8) |
| Yes | 33 (24.8) | 27 (40.9) | | 60 (30.2) |
| Nephrological, [N (%)] | | | 0.236 | |
| No | 122 (91.7) | 57 (86.4) | | 179 (89.9) |
| Yes | 11 (8.3) | 9 (13.6) | | 20 (10.1) |
| Multimorbidity, [N (%)] | | | **0.001** | |
| No | 101 (75.9) | 35 (53.0) | | 136 (68.3) |
| Yes | 32 (24.1) | 31 (47.0) | | 63 (31.7) |
| Polypharmacy, [N (%)] | | | **0.002** | |
| No | 105 (78.9) | 38 (57.6) | | 143 (71.9) |
| Yes | 28 (21.1) | 28 (42.4) | | 56 (28.1) |
| Neurosensorial deficit, [N (%)] | | | 0.062 | |
| No | 121 (91.0) | 54 (81.8) | | 175 (87.9) |
| Yes | 12 (9.0) | 12 (18.2) | | 24 (12.1) |

Data on quality of life, anxiety, depression, and HIV-related stigma are presented in **Table 5**. Across all dimensions of the EQ-5D-5L index, individuals with loneliness and/or social isolation reported significantly higher proportion of problems (p < 0.001 for each dimension). Similarly, HADS scale results indicated higher anxiety and depression levels in individuals with loneliness and/or social isolation (p < 0.001 for both). Finally, self-reported stigma, measured by the HSS Scale, was also significantly higher in the loneliness and/or social isolation group third tercile (p < 0.001).

**Table 6** shows medians (IQR) and median differences (95% CI) of inflammatory markers comparing individuals with loneliness and/or social isolation versus those with neither loneliness nor social isolation. In the unadjusted analysis, only CSF1 showed significantly elevated levels among individuals with loneliness and/or social isolation [median: 151.2 (95% CI: 134.7; 165.2)] compared to those experiencing neither [median: 139.9 (95% CI: 126.0; 153.3)], resulting in a median difference of 11.33 (95% CI 3.50; 19.15; p = 0.005). However, in the adjusted analysis, no significant differences were observed in any of the biomarkers analyzed in relation to loneliness and/or social isolation

**Table 4. Tobacco, alcohol and drug use as a function of loneliness and/or social isolation.**

| | Loneliness and/or social isolation | | | |
| --- | --- | --- | --- | --- |
| | No | Yes | p-value | Total |
| | N = 133 | N = 66 | | N = 199 |
| Have you ever been a regular tobacco user?, [N (%)] | | | 0.693 | |
| No | 27 (20.3) | 15 (22.7) | | 42 (21.1) |
| Yes | 106 (79.7) | 51 (77.3) | | 157 (78.9) |
| Risk Alcohol consumption, [N (%)] | | | 0.701 | |
| No | 82 (61.7) | 44 (66.7) | | 126 (63.3) |
| Yes | 20 (15.0) | 10 (15.2) | | 30 (15.1) |
| Unknown | 31 (23.3) | 12 (18.2) | | 43 (21.6) |
| Drug use in the last year, [N (%)] | | | 0.293 | |
| No | 100 (75.2) | 54 (81.8) | | 154 (77.4) |
| Yes | 33 (24.8) | 12 (18.2) | | 45 (22.6) |

## Discussion

In our study, we did not observe differences in the analyzed markers between individuals experiencing loneliness and/or social isolation and those without loneliness or social isolation among PLWH aged >50 years. Nonetheless, previous studies have estimated that social disconnection, including loneliness, can increase inflammation in people without HIV [23,26,55–58]. However, a systematic review and meta-analysis examining associations between loneliness, social isolation, and some inflammatory biomarkers such as C-reactive protein (CRP), fibrinogen, interleukin(IL)-1RA, IL-6, and monocyte chemotactic protein-1 (MCP-1) [25], found a significant association between loneliness and the circulating IL-6 in adjusted analyses, with no association found between social isolation and IL-6. A minimally adjusted association was observed between social isolation and the acute-phase proteins CRP and fibrinogen, though caution was advised in interpreting these findings due to heterogeneity and limited robustness in some associations [25]. Similarly, van Bogart et al. [55] reported a significantly association between loneliness and higher CRP levels, yet found but no significant associations with cytokines. These findings underscore the needed for further research to clarify these relationships. Another meta-analysis [57] reported that social support and social integration were associated with lower levels of inflammatory cytokines.

Among PLWH, Ellis et al. [24] reported a link between social isolation and inflammation, showing that lower social support was associated with elevated plasma levels of MCP-1, IL-8 and vascular endothelial growth factor (VEGF), and higher cerebrospinal fluid levels of MCP-1 and IL-6 in a combined cohort of PLWH and HIV-negative individuals. Hussain et al. [30] found that loneliness in virologically suppressed PLWH (ages 36–69 years) was associated with increased coagulation (D-dimer) and monocyte activation (sCD14, CCL2/MCP-1). In contrast, a single-site study by Derry et al. [31] of PLWH aged 54–78 years reported no significant association between loneliness and composite cytokine levels or CRP levels. Our multicenter study similarly found no significant associations, and, to our knowledge, evaluated the largest panel of cytokines to date in this population.

Loneliness is highly correlated with depression [59]. Among PLWH, greater somatic depressive symptoms were linked to a higher monocyte activation (sCD14) and altered coagulation (D-dimer) [60]. Notably, different antidepressants influence inflammation distinctly: selective serotonin reuptake inhibitors were associated with lower levels of sCD14 and IL-6 levels, while tricyclic antidepressants were linked to higher levels of these markers [60].

Although we did not observe additional inflammatory elevation among PLWH reporting loneliness or social isolation, we caution against interpreting this as evidence against an underlying association. Such a relationship may have gone undetected because our cohort, clinically stable and engaged in care, shows limited variability in psychosocial exposures and

**Table 5. Quality of life, depression and anxiety, and stigma, as a function of loneliness and/or social isolation.**

| | Loneliness and/or social isolation | | | |
| | No | Yes | p-value | Total |
| | N = 133 | N = 66 | | N = 199 |
|---|---|---|---|---|
| Quality of life – EQ-5D-5L scale | | | | |
| Any mobility issue, [N (%)] | | | **<0.001** | |
| No | 111 (83.5) | 30 (45.5) | | 141 (70.9) |
| Yes | 22 (16.5) | 36 (54.5) | | 58 (29.1) |
| Any self-care issue, [N (%)] | | | **<0.001** | |
| No | 127 (96.2) | 52 (78.8) | | 179 (90.4) |
| Yes | 5 (3.8) | 14 (21.2) | | 19 (9.6) |
| Any issue in performing usual activities, [N (%)] | | | **<0.001** | |
| No | 116 (87.9) | 37 (56.1) | | 153 (77.3) |
| Yes | 16 (12.1) | 29 (43.9) | | 45 (22.7) |
| Any pain or discomfort issue, [N (%)] | | | **<0.001** | |
| No | 88 (66.2) | 22 (33.3) | | 110 (55.3) |
| Yes | 45 (33.8) | 44 (66.7) | | 89 (44.7) |
| Any anxiety or depression issue, [N (%)] | | | **<0.001** | |
| No | 84 (63.2) | 16 (24.2) | | 100 (50.3) |
| Yes | 49 (36.8) | 50 (75.8) | | 99 (49.7) |
| Anxiety and Depression – HADS Scale | | | | |
| Clinically significant depression, [N (%)] | | | **<0.001** | |
| No | 113 (85.0) | 33 (50.0) | | 146 (73.4) |
| Yes | 18 (13.5) | 33 (50.0) | | 51 (25.6) |
| Unknown | 2 (1.5) | 0 (0.0) | | 2 (1.0) |
| Clinically significant anxiety, [N (%)] | | | **<0.001** | |
| No | 104 (78.2) | 29 (43.9) | | 133 (66.8) |
| Yes | 28 (21.1) | 37 (56.1) | | 65 (32.7) |
| Unknown | 1 (0.8) | 0 (0.0) | | 1 (0.5) |
| HIV-Stigma – HSS Scale, [N (%)] | | | **<0.001** | |
| T1/T2 | 97 (72.9) | 27 (40.9) | | 124 (62.3) |
| T3 | 31 (23.3) | 37 (56.1) | | 68 (34.2) |
| Unknown | 5 (3.8) | 2 (3.0) | | 7 (3.5) |

Note: T1: first tertile; T2: second tercile; T3: third tercile

inflammatory readouts; because exposure and biomarkers were assessed at a single time point, which may not align with the temporal dynamics of psychosocial stress and immune activation; and due to measurement or analytical constraints (e.g., limited power for small effects, residual confounding, non-linear or threshold relationships, and effect modification by demographic or clinical factors). Within this context, several biological and social factors may explain our findings: first, chronic HIV-related inflammation may approach a biological 'ceiling', blunting incremental effects of loneliness [61,62]; second; adaptation to persistent inflammation in PLWH may reduce the reactivity to psychosocial stressors, such as loneliness and SIL, limiting their additional inflammatory impact [61,62]; third supportive structures common in HIV care, such as peer support and multidisciplinary care teams, may mitigate the inflammatory impact of loneliness [63]; fourth, and HIV- and loneliness-related inflammation may involve partly distinct pathways, potentially limiting their additive effects in PLWH [64,65]. These considerations support a cautious interpretation of our findings and motivate longitudinal, multi-time-point studies with adequate power and explicit tests for non-linearity and effect modification.

**Table 6. Medians (RI) and Median Difference (95%CI) of inflammatory markers as a function of having loneliness and/or social isolation versus having neither loneliness nor social isolation. Entries are listed in alphabetical order by biomarker name.**

| | Loneliness and/or social isolation | | Crude analysis | | Adjusted analysis* | |
| --- | --- | --- | --- | --- | --- | --- |
| | No | Yes | | | | |
| | Median (IQR) | Median (IQR) | Difference in medians (yes vs. no) (95% CI) | p-value | Difference in medians (95% CI) | p-value |
| CCL11 | 190.1 (131.8 - 248.2) | 176.9 (131.9 - 239.9) | −13.04 (−39.91; 13.84) | 0.340 | 5.89 (−28.94; 40.72) | 0.739 |
| CCL13 | 174.8 (117.8 - 247.5) | 163.5 (107.6 - 241.0) | −4.42 (−45.21; 36.37) | 0.831 | −13.24 (−56.27; 29.79) | 0.545 |
| CCL19 | 121.6 (93.0 - 159.8) | 117.1 (86.3 - 154.2) | −4.42 (−23.78; 14.95) | 0.653 | −17.64 (−42.39; 7.10) | 0.161 |
| CCL2 | 625.4 (474.4 - 874.3) | 611.5 (434.7 - 765.9) | −8.41 (−109.46; 92.64) | 0.870 | −46.33 (−159.07; 66.41) | 0.419 |
| CCL3 | 9.5 (7.6 - 12.2) | 8.7 (7.1 - 10.6) | −0.67 (−1.82; 0.49) | 0.256 | −0.88 (−2.31; 0.54) | 0.222 |
| CCL4 | 129.9 (101.7 - 188.0) | 122.4 (88.5 - 160.3) | −6.83 (−32.23; 18.56) | 0.596 | 2.88 (−22.98; 28.75) | 0.826 |
| CCL7 | 1.3 (0.9 - 1.9) | 1.3 (0.9 - 1.9) | −0.02 (−0.30; 0.26) | 0.896 | −0.16 (−0.47; 0.14) | 0.296 |
| CCL8 | 66.8 (41.9 - 89.5) | 60.4 (40.6 - 83.9) | −5.54 (−17.93; 6.86) | 0.379 | −6.65 (−21.70; 8.40) | 0.385 |
| CSF1 | 139.9 (126.0 - 153.3) | 151.2 (134.7 - 165.2) | **11.33 (3.50; 19.15)** | **0.005** | 5.90 (−2.21; 14.02) | 0.153 |
| CSF3 | 117.2 (91.7 - 156.4) | 122.3 (89.8 - 147.7) | 5.57 (−10.74; 21.89) | 0.501 | 12.22 (−5.81; 30.26) | 0.183 |
| CXCL10 | 108.7 (81.6 - 171.3) | 120.0 (82.5 - 161.8) | 12.52 (−15.58; 40.63) | 0.381 | 11.88 (−19.11; 42.87) | 0.450 |
| CXCL11 | 69.4 (45.2 - 106.6) | 64.6 (46.0 - 92.5) | −4.27 (−19.77; 11.22) | 0.587 | 1.65 (−16.42; 19.72) | 0.857 |
| CXCL12 | 196.3 (171.7 - 233.1) | 208.4 (166.7 −263.0) | 13.44 (−5.65; 32.54) | 0.167 | 19.84 (−4.14; 43.82) | 0.104 |
| CXCL8 | 18.0 (13.1 - 24.7) | 16.7 (13.2 - 21.2) | −1.11 (−4.06; 1.83) | 0.457 | −1.79 (−5.87; 2.30) | 0.389 |
| CXCL9 | 88.1 (62.8 - 132.3) | 87.6 (65.6 - 125.4) | −1.06 (−19.67; 17.55) | 0.911 | −12.34 (−31.93; 7.24) | 0.215 |
| EGF | 320.3 (58.0 - 490.7) | 200.3 (42.9 - 454.1) | −100.20 (−225.57; 25.17) | 0.117 | 23.68 (−119.79; 167.14) | 0.745 |
| FLT3LG | 129.0 (107.2 - 160.5) | 131.8 (110.4 - 162.5) | 2.95 (−13.09; 18.99) | 0.717 | −4.98 (−22.83; 12.86) | 0.582 |
| HGF | 598.6 (464.8 - 792.5) | 654.9 (476.5 - 857.4) | 59.78 (−34.92; 154.49) | 0.215 | 1.80 (−120.92; 124.52) | 0.977 |
| IFNG | 0.2 (0.2 - 0.4) | 0.3 (0.2 - 0.4) | 0.01 (−0.05; 0.07) | 0.770 | −0.01 (−0.08; 0.05) | 0.711 |
| IL10 | 8.3 (5.6 - 13.9) | 8.5 (5.4 - 10.7) | 0.33 (−1.71; 2.36) | 0.752 | −1.11 (−3.37; 1.14) | 0.332 |
| IL15 | 14.4 (12.3 - 17.2) | 14.9 (13.1 - 16.9) | 0.59 (−0.62; 1.80) | 0.339 | 1.06 (−0.36; 2.48) | 0.143 |
| IL17A | 0.6 (0.3 - 1.2) | 0.5 (0.3 - 1.0) | −0.09 (−0.37; 0.19) | 0.529 | 0.02 (−0.29; 0.32) | 0.916 |
| IL17C | 23.9 (16.8 - 38.1) | 23.6 (18.5 - 34.8) | 0.06 (−6.08; 6.21) | 0.983 | 1.59 (−4.19; 7.38) | 0.587 |
| IL17F | 0.7 (0.4 - 1.2) | 0.6 (0.4 - 1.4) | −0.16 (−0.38; 0.05) | 0.131 | −0.09 (−0.33; 0.15) | 0.454 |

*(Continued)*

**Table 6.** (Continued)

| | Loneliness and/or social isolation | | Crude analysis | | Adjusted analysis* | |
| | No | Yes | | | | |
| | Median (IQR) | Median (IQR) | Difference in medians (yes vs. no) (95% CI) | p-value | Difference in medians (95% CI) | p-value |
|---|---|---|---|---|---|---|
| IL18 | 352.7 (284.5 - 473.8) | 375.8 (295.4 - 503.0) | 23.31 (−25.07; 71.70) | 0.343 | 34.83 (−29.85; 99.52) | 0.289 |
| IL27 | 9.2 (5.2 - 14.0) | 9.0 (6.0 - 12.1) | −0.16 (−2.52; 2.20) | 0.893 | 1.38 (−1.31; 4.07) | 0.314 |
| IL6 | 3.4 (2.5 - 5.9) | 3.7 (2.5 - 6.2) | 0.31 (−0.59; 1.21) | 0.499 | −0.50 (−1.52; 0.52) | 0.337 |
| IL7 | 5.9 (4.0 - 7.8) | 5.8 (3.9 - 7.6) | −0.10 (−1.22; 1.02) | 0.863 | −0.51 (−1.62; 0.60) | 0.367 |
| LTA | 8.5 (7.2 - 9.7) | 8.6 (7.2 - 10.3) | 0.16 (−0.52; 0.84) | 0.641 | −0.09 (−0.94; 0.76) | 0.834 |
| MMP1 | 2462.3 (1366.4 - 4011.5) | 2963.1 (1455.4- 4233.9) | 579.48 (−188.15;1347.11) | 0.138 | −43.69 (−893.62; 806.25) | 0.919 |
| MMP12 | 349.8 (275.8 - 486.7) | 393.5 (256.9 - 573.2) | 61.96 (−8.24; 132.16) | 0.083 | 23.74 (−50.64; 98.12) | 0.530 |
| OLR1 | 312.2 (156.2 - 488.0) | 277.2 (132.9 - 530.1) | −31.31 (−130.86; 68.25) | 0.536 | 70.38 (−60.58; 201.34) | 0.290 |
| OSM | 7.8 (5.0 - 11.6) | 7.9 (3.9 - 12.5) | 0.43 (−1.54; 2.40) | 0.667 | 1.26 (−0.96; 3.48) | 0.264 |
| TGFA | 18.1 (12.4 - 25.9) | 19.6 (11.4 - 31.9) | 1.61 (−2.91; 6.12) | 0.483 | 1.53 (−3.69; 6.75) | 0.564 |
| TNF | 19.7 (16.2 - 23.9) | 18.8 (15.9 - 22.4) | −0.78 (−2.90; 1.34) | 0.469 | −0.69 (−3.41; 2.04) | 0.620 |
| TNFSF10 | 505.6 (432.4 - 579.6) | 489.4 (419.3 - 554.3) | −15.52 (−48.65; 17.62) | 0.357 | −4.36 (−53.31; 44.58) | 0.861 |
| TNFSF12 | 724.2 (599.6 - 890.3) | 697.3 (602.5 - 907.1) | −26.83 (−99.60; 45.95) | 0.468 | 21.79 (−62.84; 106.42) | 0.612 |
| VEGFA | 606.0 (404.1 - 947.8) | 528.8 (371.7 - 928.2) | −65.87 (−208.89; 77.16) | 0.365 | −130.32 (−302.93; 42.30) | 0.138 |

*Adjusted for sex at birth (male, female), age (50–59, 60 + years), employment status (student/worker, retiree/unemployed, unknown), time since HIV diagnosis (<10, 10–19, ≥ 20 years), current CD4/CD8 ratio (≤1, > 1), current HIV RNA < 50 copies/ml (no, yes), multimorbidity (no, yes), polypharmacy (no, yes), ever regular tobacco use (no, yes), and clinically significant depression (no, yes, unknown) and anxiety (no, yes, unknown).

Several limitations should be considered in interpreting these results. First, it is observational, so we cannot infer causality or tell which came first. Second, contrary to our expectations, were found no association between loneliness/social isolation and systemic inflammation; this may reflect unmeasured confounding that influences both social support and inflammation [24]. Third, effects may be non-linear, with stronger associations among those most lonely. Fourth, the use of internationally accepted brief screening questionnaires for loneliness and SIL [66–76], which, for example, do not assess duration, may have limited the detection of inflammation changes associated with prolonged exposure, whereas measures capturing such dimensions could reveal stronger associations with inflammatory markers. Fifth, our study did not include an HIV-negative control group because our focus was mechanisms within PLWH. Sixth, while systemic inflammation is essential for understanding chronic diseases, current measurement methods vary in cost, utility, and predictive validity [23]. Biomarkers employed in this study were those associated with HIV-related disease processes, yet emerging biomarkers (i.e., microbiota, soluble urokinase plasminogen activator receptor) may offer more reliable indicators of chronic inflammation than traditional markers like high-sensitivity CRP and IL-6 [23,77]. Chronic rather than acute inflammatory

biomarkers may be particularly relevant to loneliness and/or social isolation research [23]. Seventh, loneliness may be a consequence of psychosocial stressors, such as HIV-related stigma [25]. Finally, non-ART factors not accounted here could potentially impact on the inflammatory state [60]. These points argue for longitudinal, mechanistic studies with harmonised measurement; despite them, our findings offer clinically relevant insight.

In summary, our findings contribute to the growing body of linking loneliness with inflammation. However, this study did no observe a significant increase in inflammatory markers among PLWH experiencing loneliness and/or social isolation. Further studies are needed to confirm these findings and to explore other potential factors that may influence these outcomes.

## Supporting information

**S1 Table. Detailed information on the Olink Target 48 Cytokine panel.**
(XLS)

## Acknowledgments

**Other researchers from the Gesida 12021 study group**

 Hospital Clínico Universitario Lozano Blesa, Zaragoza (MJ. Crusells Canales; S. Letona Carbajo).

 Hospital de Viladecans – Institut Català de la Salut (C. Imperiali-Rosario; M. Ruiz-Pombo)

 Hospital General Universitario Dr Balmis – Instituto de Investigación Sanitaria y Biomédica de Alicante (ISABIAL) (J. Portilla Sogorb; E. Merino de Lucas; V. Boix Martínez; D. Torrús Tendero; S. Reus Bañuls, G. García Rodríguez; L. Paredes Arquiola; L. Giner Oncina; I. Agea Durán)

 Hospital Universitario Clínico San Carlos, Madrid (V. Estrada Pérez)

 Hospital Universitario de La Princesa, Madrid (L. García-Fraile)

 Hospital Universitario La Paz, IdiPAZ, Madrid (M.L. Montes Ramírez; A. Delgado Hierro)

 Hospital Universitario Río Hortega, Valladolid (B. Valentin Casado; M. González Fernández; M. Cazorla González; J. Gómez Barquero; RM. Lobo Valentin; Mª del Carmen Rebollo-Nájera)

 Hospital Universitario San Pedro (E. Melús; M. Barrios)

 Hospital Universitario Severo Ochoa, Leganés, Madrid (M. Cervero Jiménez; C. García-Lacalle; C. Córdoba-Chicote)

 Hospital Universitario Virgen de la Arrixaca, Murcia (A. Castillo Navarro)

## Author contributions

**Conceptualization:** Jose-Ramon Blanco, Javier de la Torre, Alicia González-Baeza, Herminia Esteban, Julian Olalla.

**Data curation:** Inma Jarrin.

**Formal analysis:** Lourdes Romero, Laura Perez-Martinez, Inma Jarrin.

**Funding acquisition:** Jose-Ramon Blanco, Javier de la Torre, Alicia González-Baeza.

**Investigation:** Jose-Ramon Blanco, Helena Albendin-Iglesias, Ana Mª Barrios-Blandino, Cristina Tomás-Jimenez, Isabel Sanjoaquin-Conde, María Saumoy, Verónica Pérez-Esquerdo, Inmaculada Gonzalez-Cuello, Ana María López-Lirola, María José Galindo, Noemí Cabello-Clotet, Jesica Abadía Otero, Dolores Merino-Muñoz, Joanna Cano-Smith, Magdalena Muelas-Fernandez, Javier de la Torre, Alicia González-Baeza, Lourdes Romero, Antonio Ocampo, Rafael Torres, Carmen Hidalgo, Maria Angeles Fernandes-López, Jordi Puig, Lucio Garcia Fraile, Enrique Bernal Morell, Laura Perez-Martinez, Julian Olalla.

**Methodology:** Jose-Ramon Blanco, Javier de la Torre, Alicia González-Baeza, Herminia Esteban, Laura Perez-Martinez, Marta De Miguel Montero, Inma Jarrin, Julian Olalla.

**Project administration:** Herminia Esteban, Marta De Miguel Montero.

**Resources:** Herminia Esteban, Marta De Miguel Montero.

**Supervision:** Jose-Ramon Blanco, Helena Albendin-Iglesias, Eugenia Negredo-Puigmal, Ana Mª Barrios-Blandino, Cristina Tomás-Jimenez, Isabel Sanjoaquin-Conde, María Saumoy, Verónica Pérez-Esquerdo, Inmaculada Gonzalez-Cuello, Ana María López-Lirola, María José Galindo, Noemí Cabello-Clotet, Jesica Abadía Otero, Dolores Merino-Muñoz, Joanna Cano-Smith, Magdalena Muelas-Fernandez, Antonio Ocampo, Rafael Torres, Carmen Hidalgo, Maria Angeles Fernandes-López, Jordi Puig, Lucio Garcia Fraile, Enrique Bernal Morell, Julian Olalla.

**Validation:** Eugenia Negredo-Puigmal, Ana Mª Barrios-Blandino, Cristina Tomás-Jimenez, Isabel Sanjoaquin-Conde, María Saumoy, Verónica Pérez-Esquerdo, Inmaculada Gonzalez-Cuello, Ana María López-Lirola, María José Galindo, Noemí Cabello-Clotet, Jesica Abadía Otero, Dolores Merino-Muñoz, Joanna Cano-Smith, Magdalena Muelas-Fernandez, Antonio Ocampo, Rafael Torres, Carmen Hidalgo, Maria Angeles Fernandes-López, Jordi Puig, Lucio Garcia Fraile, Enrique Bernal Morell, Laura Perez-Martinez, Inma Jarrin.

**Visualization:** Eugenia Negredo-Puigmal, Ana Mª Barrios-Blandino, Cristina Tomás-Jimenez, Isabel Sanjoaquin-Conde, María Saumoy, Verónica Pérez-Esquerdo, Inmaculada Gonzalez-Cuello, Ana María López-Lirola, María José Galindo, Noemí Cabello-Clotet, Jesica Abadía Otero, Dolores Merino-Muñoz, Joanna Cano-Smith, Magdalena Muelas-Fernandez, Antonio Ocampo, Rafael Torres, Carmen Hidalgo, Maria Angeles Fernandes-López, Jordi Puig, Lucio Garcia Fraile, Enrique Bernal Morell.

**Writing – original draft:** Jose-Ramon Blanco, Javier de la Torre, Alicia González-Baeza, Inma Jarrin, Julian Olalla.

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
