## [Decision Letter · Decision Letter 0]

15 Aug 2025

Dear Dr. Blanco,

Thank you for submitting your manuscript to PLOS ONE. After careful consideration, we feel that it has merit but does not fully meet PLOS ONE’s publication criteria as it currently stands. Therefore, we invite you to submit a revised version of the manuscript that addresses the points raised during the review process.

We look forward to receiving your revised manuscript.

Kind regards,

Stanley Chinedu Eneh

Academic Editor

PLOS ONE

Journal Requirements:

Additional Editor Comments:

The measurement of inflammatory biomarkers is not clearly described. It is important to specify the methods used for measuring these biomarkers.Provide the ethical approval number for this studyProvide sample size justification

Kindly revise the manuscript and provide answers per the comments provided by the reviewer(s).

Reviewer's Responses to Questions

**Comments to the Author**

1. Is the manuscript technically sound, and do the data support the conclusions?

Reviewer #1: Yes

Reviewer #2: No

Reviewer #3: Partly

2. Has the statistical analysis been performed appropriately and rigorously?

Reviewer #1: Yes

Reviewer #2: Yes

Reviewer #3: No

3. Have the authors made all data underlying the findings in their manuscript fully available?

Reviewer #1: No

Reviewer #2: Yes

Reviewer #3: Yes

4. Is the manuscript presented in an intelligible fashion and written in standard English?

Reviewer #1: Yes

Reviewer #2: Yes

Reviewer #3: Yes

Reviewer #1: In this study, the authors analysed data from 199 people living with HIV and observed that participants who experienced loneliness/social isolation did not show differences in peripheral inflammatory profiles compared to those who did not.

While this is a solid study with a well-considered methodology, I have a few suggestions and questions to improve the manuscript:

1. Please stick to consistent and accepted person-first terminology: “people with HIV” and “people without HIV”, not “HIV positive” or “HIV negative”.

2. Please clarify in the methods whether the questionnaires and the blood sampling was carried out within the same study visit, or if there was a time lag between these measurements.

3. Table 8 would be more readable if the inflammatory biomarkers were arranged alphabetically, or in some other logical order. At the moment, the order seems fairly random.

4. The discussion could benefit from citing some more relevant and recent reviews (in addition to individual studies), for instance see PMC10489482.

5. The authors have provided some reasonable explanations for why they may not have observed significant associations between inflammation and loneliness/social isolation in their study. However, I would like to see a more comprehensive discussion of this, since the association between these factors has been reported fairly consistently in several studies and even meta-analyses in the general population (which the authors have cited) as well as some individual studies in people with HIV. The potential explanations provided in the discussion at the moment assume that there is no true underlying association, and not the alternative explanation that the underlying relationship has simply not been captured in the current study, which may be due to (for instance) the study population characteristics or the single time-point measurement. Please discuss in more detail.

6. Finally, there is no justification offered for the lack of data availability. It is a growing expectation (and indeed standard policy for the specific journal where the manuscript has been submitted) that data is made publicly available in de-identified form. I am not sure why data has not been made available in this case.

Reviewer #2: The concepts of loneliness and social isolation are related but distinct. The authors need to describe each individually in the background and how they differ both in conceptualization and operationalization. The relationship between loneliness and inflammation has been described elsewhere and the background would benefit from an overview of this work and identifying the relationship between loneliness and inflammation on the causal pathway to other health outcomes.

The authors need to describe the variables included in the analysis in the Methods. Although this might be relevant to another published study this information is needed tin text to provide context for the information provided in the tables and results.

The authors describe the cut-offs (>6 3-item scale) as the presence of loneliness and isolation but what they are actually describing is the severity of loneliness. Relevant articles using similar cut-offs should be cited and the rationale for >6 (moderate loneliness) versus >7 (severe loneliness) should be included.

It was challenging to interpret any of the results because of the mixing of loneliness and social isolation. The tables and reporting in the Results currently mix both concepts and they should be described distinctly as they are distinct concepts. Given the mixing of the two main concepts it is challenging to meaningfully interpret the findings.

Reviewer #3: <overall evaluation=" ">

This study investigates the association between loneliness/social isolation and chronic inflammation in older people living with HIV (PLWH). Its multi-centre design, use of standardised rating scales, and comprehensive cytokine panel are noteworthy. The principal finding—that no clear association was detected—is itself valuable. The manuscript’s overall structure is appropriate, and the methodology appears sound. However, several issues, chiefly regarding the statistical analyses and the interpretation of results, require revision. Addressing these points will clarify the study’s scientific value and strengthen its suitability for publication in PLOS ONE.

<specific comments=" ">

Abstract

The concluding sentence (“Further research is needed to validate these findings and explore additional factors”) is rather general. Briefly touching on specific future research directions suggested by your results would be more helpful to readers.

Introduction

The research objective is clearly stated, but the authors do not present an explicit hypothesis (e.g., “PLWH with greater loneliness or social isolation will show higher levels of specific inflammatory markers than those without such experiences”). Stating a directional hypothesis would sharpen the study’s focus and guide interpretation of the results.

Materials and Methods

Participants and Data

Key details on participant selection/exclusion criteria and the handling of missing values should be included in the main text rather than referenced only in earlier studies.

Measurements

The study was conducted in Spain, yet the manuscript does not specify whether English or Spanish versions of the UCLA 3-item Loneliness Scale and the Lubben Social Network Scale-6 were used. If Spanish versions were employed, please cite validation studies and report their internal consistency (e.g., Cronbach’s α).

The rationale for the cut-offs (“UCLA ≥ 6” and “Lubben ≤ 20”) should also be provided.

Regarding inflammatory markers, the manuscript excludes markers for which more than 30 % of values fell below the detection limit (IL-33, IL-2, IL-1β, IL-4, TSLP, IL-13, CSF-2). Please explain why the 30 % threshold was chosen.

Statistical Analysis

The adjustment variables are described as “variables commonly associated with inflammatory markers.” A more detailed justification—supported by references—would help readers understand why each variable was considered a potential confounder. If the number of covariates is large relative to the sample size, model over-fitting or reduced power may result; please discuss this risk.

In addition to p-values, please report effect-size measures; these help readers judge clinical relevance regardless of statistical significance.

Results

Participant Characteristics

Several discrepancies exist between the narrative text and Table 1:

The text states that individuals without loneliness/isolation were more likely to be women (p = 0.023), yet Table 1 shows a higher proportion of men (77.4 %) in the “No” group and more women (37.9 %) in the “Yes” group.

Similarly, the text says that those without loneliness/isolation were more likely to be unemployed or retired (p < 0.001), whereas Table 1 indicates the opposite (higher employment in the “No” group).

The text references “…living alone unwillingly (p = 0.006)”, but this item is absent from Table 1.

Please revise the prose, correct the table, or add the missing variable as appropriate.

Multiple Comparisons

Across Tables 1–5, numerous variables are compared, generating many p-values. The risk of false-positive findings due to multiple testing should be acknowledged, and some form of adjustment (e.g., modified significance level or FDR control) or cautious interpretation is warranted. Reporting effect sizes would also help.

Discussion

Comparison with previous research needs to go beyond listing earlier findings. Where results diverge, discuss plausible reasons (e.g., differences in sample characteristics, loneliness measures, inflammatory marker assays, or statistical methods) and position your study accordingly (supporting, contradicting, or refining previous work). This will clarify both the contribution and the limitations of the present study.

<conclusion>

The finding of no clear association between loneliness/social isolation and inflammatory markers in PLWH is scientifically valuable. By clarifying the methodology, refining the statistical analysis, ensuring consistency in the presentation of results, and deepening the discussion, the manuscript’s quality—and its contribution to the field—will be further enhanced. I look forward to reviewing a revised version.</conclusion></specific></overall>

**Do you want your identity to be public for this peer review?** For information about this choice, including consent withdrawal, please see our Privacy Policy

Reviewer #1: No

Reviewer #2: No

Reviewer #3: No

---

## [Author Response · Author response to Decision Letter 1]

17 Sep 2025

POINT-BY-POINT RESPONSE TO THE REFEREES’ COMMENTS

Reviewer #1:

1. Please stick to consistent and accepted person-first terminology: “people with HIV” and “people without HIV”, not “HIV positive” or “HIV negative”.

Thank you for your suggestion. We have updated the manuscript to ensure consistent use of person-first language throughout.

2. Please clarify in the methods whether the questionnaires and the blood sampling was carried out within the same study visit, or if there was a time lag between these measurements.

Thank you for this helpful comment. We have clarified in Methods that the questionnaires and blood sampling were conducted as part of the same study.

3. Table 8 would be more readable if the inflammatory biomarkers were arranged alphabetically, or in some other logical order. At the moment, the order seems fairly random.

Thank you for the suggestion. We understand the comment refers to Table 6, as there is no Table 8 in the manuscript. We have done it.

4. The discussion could benefit from citing some more relevant and recent reviews (in addition to individual studies), for instance see PMC10489482.

Thank you for your suggestion. We have added that citation and a few others.

5. The authors have provided some reasonable explanations for why they may not have observed significant associations between inflammation and loneliness/social isolation in their study. However, I would like to see a more comprehensive discussion of this, since the association between these factors has been reported fairly consistently in several studies and even meta-analyses in the general population (which the authors have cited) as well as some individual studies in people with HIV. The potential explanations provided in the discussion at the moment assume that there is no true underlying association, and not the alternative explanation that the underlying relationship has simply not been captured in the current study, which may be due to (for instance) the study population characteristics or the single time-point measurement. Please discuss in more detail.

Thank you for highlighting this point. We have revised the Discussion to acknowledge that, beyond explanations consistent with a true null additive effect, an underlying relationship may exist but was not identified.

6. Finally, there is no justification offered for the lack of data availability. It is a growing expectation (and indeed standard policy for the specific journal where the manuscript has been submitted) that data is made publicly available in de-identified form. I am not sure why data has not been made available in this case.

Thank you for raising this point. Individual-level data from human participants cannot be made publicly available due to privacy and informed-consent restrictions. De-identified data will be available to qualified researchers upon reasonable request and completion of a data-use agreement, subject to approval by the Institutional Research Ethics Committee (Comité de Ética de Investigación con medicamentos de La Rioja [CEImLAR]).

Reviewer #2:

1. The concepts of loneliness and social isolation are related but distinct. The authors need to describe each individually in the background and how they differ both in conceptualization and operationalization. The relationship between loneliness and inflammation has been described elsewhere and the background would benefit from an overview of this work and identifying the relationship between loneliness and inflammation on the causal pathway to other health outcomes.

Thank you for your comment. We have done it.

2. The authors need to describe the variables included in the analysis in the Methods. Although this might be relevant to another published study this information is needed tin text to provide context for the information provided in the tables and results.

Thank you for your comment. We have done it.

3. The authors describe the cut-offs (>6 3-item scale) as the presence of loneliness and isolation but what they are actually describing is the severity of loneliness. Relevant articles using similar cut-offs should be cited and the rationale for >6 (moderate loneliness) versus >7 (severe loneliness) should be included.

Thank you for your comment. The cut-offs for the 3-item UCLA scale is described and justified in the Material and Method section.

4. It was challenging to interpret any of the results because of the mixing of loneliness and social isolation. The tables and reporting in the Results currently mix both concepts and they should be described distinctly as they are distinct concepts. Given the mixing of the two main concepts it is challenging to meaningfully interpret the findings.

Thank you for raising this important point. We agree that loneliness and social isolation are distinct constructs and that combining them can obscure interpretation. Initially, we treated them as separate exposures and classified participants into four mutually exclusive groups (neither, loneliness only, social isolation only, both). Because some strata were small and no meaningful differences in inflammatory markers were observed among the loneliness only, social isolation only, and both groups, we combined these categories into a single exposure (‘loneliness and/or social isolation’) to enhance statistical power. We have revised the Results section to clarify this.

Reviewer #3:

1. Abstract: The concluding sentence (“Further research is needed to validate these findings and explore additional factors”) is rather general. Briefly touching on specific future research directions suggested by your results would be more helpful to readers.

Thank you for your comment. We have revised it.

2. Introduction: The research objective is clearly stated, but the authors do not present an explicit hypothesis (e.g., “PLWH with greater loneliness or social isolation will show higher levels of specific inflammatory markers than those without such experiences”). Stating a directional hypothesis would sharpen the study’s focus and guide interpretation of the results.

Thank you for your suggestion. We have included it.

3. Materials and Methods: Participants and Data Key details on participant selection/exclusion criteria and the handling of missing values should be included in the main text rather than referenced only in earlier studies.

Thank you for your comment. We have revised it.

4. Measurements: The study was conducted in Spain, yet the manuscript does not specify whether English or Spanish versions of the UCLA 3-item Loneliness Scale and the Lubben Social Network Scale-6 were used. If Spanish versions were employed, please cite validation studies and report their internal consistency (e.g., Cronbach’s α). The rationale for the cut-offs (“UCLA ≥ 6” and “Lubben ≤ 20”) should also be provided.

Regarding inflammatory markers, the manuscript excludes markers for which more than 30 % of values fell below the detection limit (IL-33, IL-2, IL-1β, IL-4, TSLP, IL-13, CSF-2). Please explain why the 30 % threshold was chosen.

Thank you for your observation. We used the validated Spanish versions. We have added the corresponding validation citations and report internal consistency in our study. The cut-offs for UCLA and Lubben are described and justified in the Material and Method section. Regarding inflammatory markers, we set the analyte-level threshold at 30% to balance data retention with reliability. First, this mirrors common practice in large Olink cohorts, where proteins are typically dropped once missingness/sub-LOD approaches one-third of observations (i.e., Walker et al., 2024; You et al., 2023). Second, 30% is a conservative choice within the widely used 30–50% range and helps avoid unstable inference under heavy left-censoring. Third, Olink recommends using detectability as a QC metric; requiring ≥70% detectable values provide a simple, platform-aware rule. For markers that passed this filter, sub-LOD values were imputed at the LOD to preserve information while minimizing downward bias. We now state these decisions explicitly in the Methods and cite the relevant sources.

5. Statistical Analysis: The adjustment variables are described as “variables commonly associated with inflammatory markers.” A more detailed justification—supported by references—would help readers understand why each variable was considered a potential confounder. If the number of covariates is large relative to the sample size, model over-fitting or reduced power may result; please discuss this risk. In addition to p-values, please report effect-size measures; these help readers judge clinical relevance regardless of statistical significance.

Thank you for this request. Measures of association (ie. differences in medians) were calculated to address the main objective of the study, evaluating whether inflammatory markers differed according to loneliness and/or social isolation, as shown in Table 6. We have added an explicit, literature-based rationale for covariate selection in the Statistical Analysis section.

6. Results: Participant Characteristics. Several discrepancies exist between the narrative text and Table 1: The text states that individuals without loneliness/isolation were more likely to be women (p = 0.023), yet Table 1 shows a higher proportion of men (77.4 %) in the “No” group and more women (37.9 %) in the “Yes” group. Similarly, the text says that those without loneliness/isolation were more likely to be unemployed or retired (p < 0.001), whereas Table 1 indicates the opposite (higher employment in the “No” group). The text references “…living alone unwillingly (p = 0.006)”, but this item is absent from Table 1. Please revise the prose, correct the table, or add the missing variable as appropriate.

Thank you for this helpful observation. We have corrected the text and added the missing data

7. Multiple Comparisons: Across Tables 1–5, numerous variables are compared, generating many p-values. The risk of false-positive findings due to multiple testing should be acknowledged, and some form of adjustment (e.g., modified significance level or FDR control) or cautious interpretation is warranted. Reporting effect sizes would also help.

Thank you for your comment. The sole purpose of Tables 1–5 was to characterize participants according to the presence or absence of loneliness and/or social isolation and to assess differences in these characteristics. While these tables focus on differences in percentages rather than statistical significance, measures of association were calculated to address the main objective of the study—evaluating whether inflammatory markers differed according to loneliness and/or social isolation, as shown in Table 6.

8. Discussion: Comparison with previous research needs to go beyond listing earlier findings. Where results diverge, discuss plausible reasons (e.g., differences in sample characteristics, loneliness measures, inflammatory marker assays, or statistical methods) and position your study accordingly (supporting, contradicting, or refining previous work). This will clarify both the contribution and the limitations of the present study.

Thank you for your suggestion. We have included it.

---

## [Decision Letter · Decision Letter 1]

19 Oct 2025

Lack of Association Between Loneliness, Social Isolation and Inflammation in People Living with HIV Aged ≥50 Years: Results from the Sub-Study “No One Alone-Gesida Study”.

PONE-D-25-03925R1

Dear Dr. Blanco,

We’re pleased to inform you that your manuscript has been judged scientifically suitable for publication and will be formally accepted for publication once it meets all outstanding technical requirements.

Kind regards,

Stanley Chinedu Eneh

Academic Editor

PLOS ONE

Additional Editor Comments (optional):

Reviewers' comments:

Reviewer's Responses to Questions

**Comments to the Author**

Reviewer #1: All comments have been addressed

Reviewer #3: All comments have been addressed

2. Is the manuscript technically sound, and do the data support the conclusions?

Reviewer #1: (No Response)

Reviewer #3: Yes

3. Has the statistical analysis been performed appropriately and rigorously?

Reviewer #1: (No Response)

Reviewer #3: Yes

4. Have the authors made all data underlying the findings in their manuscript fully available?

Reviewer #1: (No Response)

Reviewer #3: Yes

5. Is the manuscript presented in an intelligible fashion and written in standard English?

Reviewer #1: (No Response)

Reviewer #3: Yes

Reviewer #1: (No Response)

Reviewer #3: The authors have responded to the reviewers’ comments sincerely and appropriately, and no particular issues are identified with the revised manuscript for publication.

**Do you want your identity to be public for this peer review?** For information about this choice, including consent withdrawal, please see our Privacy Policy

Reviewer #1: No

Reviewer #3: No

---

## [Editor Report · Acceptance letter]

PONE-D-25-03925R1

PLOS One

Dear Dr. Blanco,

I'm pleased to inform you that your manuscript has been deemed suitable for publication in PLOS One. Congratulations! Your manuscript is now being handed over to our production team.

Kind regards,

on behalf of

Dr. Stanley Chinedu Eneh

Academic Editor

PLOS One